# Conceptualizing “Family” and the Role of “Chosen Family” within the LGBTQ+ Refugee Community: A Text Network Graph Analysis

**DOI:** 10.3390/healthcare9040369

**Published:** 2021-03-25

**Authors:** Seohyun Kim, Israel Fisseha Feyissa

**Affiliations:** Department of Social Welfare, Jeonbuk National University, Jeonju 54888, Korea; sh_kim@jbnu.ac.kr

**Keywords:** chosen family, family, LGBTQ+ refugees, text network graph analysis, meaning conceptualization, wellbeing and sexuality, social inclusion and sense of community

## Abstract

This study analyzed meaning attributions regarding “family” and “chosen family” by Lesbian, Gay, Bisexual, Pansexual, Transgender, Gender Queer, Queer, Intersex, Agender, Asexual, and other Queer-identifying community (LGBTQ+) refugees. The meaning and significance of a chosen family in the newly established life of the refugees was also pin-pointed for its value of safekeeping the wellbeing and settlement process. We analyzed narrative statements given by 67 LGBTQ+ refugees from 82 YouTube videos. Using InfraNodus, a text graph analysis tool, we identified pathways for meaning circulation within the narrative data, and generated a contextualized meaning for family and chosen family. The conceptualization process produced a deduction within family relationships, exploring why people, other than in biological relationships, appear to be vital in their overall wellbeing and settlement, as well as the process through which this occurs. Biological family is sometimes associated with words that instigate fear, danger, and insecurity, while the concept of chosen family is associated with words like trusting, like-minded, understanding, welcoming, loving, committed, etc. The results of the study are intended to add knowledge to the gap by showing the types and characteristics of family relationships in LGBTQ+ refugee settings. It is also a call for the relevant research community to produce more evidence in such settings, as this is essential for obtaining a better understanding of these issues.

## 1. Introduction

Across different cultures, the nuclear family is traditionally defined in a way that excludes Lesbian, Gay, Bisexual, Pansexual, Transgender, Gender Queer, Queer, Intersex, Agender, Asexual, and other Queer-identifying community (LGBTQ+) family members. The umbrella term LGBTQ+ could encompass sexual and gender minorities who identify as such, or those who question their sexual identity or who engage in a sexual act or a romantic relationship with same-sex/-gender partners [1,2]. LGBTQ+ is a common abbreviation for the Lesbian, Gay, Bisexual, Pansexual, Transgender, Gender Queer, Queer, Intersex, Agender, Asexual, and other Queer-identifying community. Exclusive sexual identification as non-heterosexual also helps to demarcate the identity of LGBTQ+ individuals [3]. The traditional family dynamics are often contested if an individual who identifies themselves as LGBTQ+ goes against the family norm. For instance, in some cases, an LGBTQ+ individual from a traditional Christian or Muslim cultural background could experience complete family disownment. Within most cultural ideals, generational continuity, family honor, and culturally honorable types of sexuality are predetermined, and LGBTQ+ individuals are either forced to conform by hiding their identity or have to flee after the disclosure of their sexuality. Although there might be a natural tendency to want to maintain their relationship with their family while expressing their sexual identity, it is also common for most families to resort to disowning as a result of culturally held beliefs. These exclusions, on many occasions, have led many LGBTQ+ people to seek “alternative families” or “families of choice” that offer them the love and security that they did not receive from their biological families [4].

Within studies about the LGBTQ+ population, the profound impact of the family of the LGBTQ+ person on the LGBTQ+ person’s life is often mentioned briefly rather than explored further [5]. For instance, in the general LGBTQ+ population, parent and family rejection is strongly associated with mental health problems, substance use, and sexual risk, while perceived family support is associated with better mental health and less substance use [6,7,8,9,10,11,12,13,14]. Nonetheless, these findings still fall short to explain which factors contribute to resilience among LGBTQ+ youth with unsupportive or rejecting families, how the presence of one supportive parent can compensate for a lack of support from another parent or guardian, how the presence of a non-parental family member can compensate for the effects of unsupportive parents, how non-parental mentors improve health outcomes, etc. [5].

Recent research findings also suggest that opposition to homosexuality and same-sex relationships in recent years has softened for the general population [15,16]. However, these types of studies need cautious interpretation [17] and should also include the subcultural characteristics of different types of LGBTQ+ individuals. One focus of this study will be to take a closer look at the sub-characteristics of the LGBTQ+ refugees who are in exile mainly because of their sexual orientation. The family dynamics in LGBTQ+ refugees are different because, compared with the general LGBTQ+ population, LGBTQ+ refugees face additional challenges due to their refugee status or minority status. These challenges will thus contest the assumption that there has been a softening in opposition towards homosexuality and same-sex relationships for minorities.

### 1.1. The Family Dynamics within LGBTQ+ Refugees

According to the United Nations High Commissioner for Refugees (UNHCR), the number of forcefully displaced individuals around the world has grown dramatically from 33.9 million in 1997 to 79.5 million at the end of 2019. Within this population, the number of LGBTQ+ refugees who are forced to flee their homes is also reported to have had a dramatic increase [18]. The term LGBTQ+ refugees refers to individuals who are forced to flee their country because persecution solely due to their gender identity, or due to sexual expression that differs from the societal norms.

Homosexuality continues to be criminalized in 72 countries worldwide, with several countries—or non-state actors within the country, such as the Islamic State in Iraq and Syria—imposing the death penalty for same-sex relations. In addition, in some of the 120 countries where homosexuality has been decriminalized, LGBTQ+ people often face severe social stigma and regular persecution. Some of the reasons many LGBTQ+ refugees have fled their home countries and sought refuge overseas include death threats, home vandalization, beatings, forced gay cures, etc. In most of these scenarios, it is the immediate family or one’s community inflicting these dangers [19,20,21].

During the exile and resettlement process, studies on LGBTQ+ refugees report that these refugees reported a lack of connection with their diaspora communities [21], more or continued hostility from the exiled country [22], a psychologically damaging burden of proof that they are members of a sexual or gender minority group [23,24,25], systematic homo/bi/trans phobia and racist systems [22,26], higher rates of refugee status refusal compared with other groups [27,28], resettlement inadequacies such as a lack of cultural competence [29], and various mental health stresses due to their minority status [17].

Within the study of refugees and asylum seekers, the issue of refugees who fled their country because of discrimination against their sexual orientation has its own dynamics, and it would be a mistake to equate their refugee experiences with the non-LGBTQ+ refugees fleeing persecution. For instance, leaving one’s country or separation from a family, for most LGBTQ+ refugees, is not always necessarily painful. In most cases, these refugees are fleeing a homophobic culture, punishing laws, or their own family members. For most of them, persecution starts within their family [19], which will, of course, be reinforced by unaccepting society and social rules. Thus, fleeing from their family and that type of society usually produces a feeling of relative relief for the refugee.

In most countries LGBTQ+ refugees flee from, it is typical for multiple generations of families to live together, sharing the same set of cultural values and leaving few opportunities for independence or a private expression of sexuality. Their early experiences, often pre-migration, indicate different forms of abuse, harassment, and threats from parents, peers, and the larger community surrounding them [19,20,21]. Consequentially, for most LGBTQ+ refugees, the notion of family remains an unstructured utopia, as in a traditional family, values are often used to justify the exclusion of LGBTQ+ daughters and sons from their families, communities, and from legal protection. These particular types of persecutions are reported as factors for LGBTQ+ refugees having difficulty placing trust in newly established support systems [21,30].

However, given the relationship dynamics LGBTQ+ refugees might have with their families, relevant studies investigating this gap are scarce. In light of poor family dynamics as the primary reason for LGBTQ+ refugees fleeing their country, conceptualization of what “family” is to LGBTQ+ refugees demands academic investigation.

### 1.2. Alternative Families: A Chosen Family

A chosen family is a group of individuals who deliberately choose one another to play significant roles in each other’s lives. One way of understanding a chosen family is as a group of people to whom a person is emotionally close enough to consider them as family, even though they are not biologically or legally related [4,31]. because of shared circumstances, chosen families in the LGBTQ+ population are considered more emotionally and psychologically supportive than biological families [32]. Families created outside of biological families are covered in the studies of Mitchell [33], Rodriguez [34], and Weston [35]. Within these studies, the traditional concept of what a family should be is alternatively constructed as more fitting to the needs of these communities. Within chosen families, the notion of “family” involves a responsibility to guarantee and commitment, as well as a willingness, to providing comfort in times of trouble [33].

In Weeks et al. [4], chosen families are defined as highly committed and friendship-based kin-like relationships. The typical characteristic of these chosen families is that they are “actively created as a positive step to reinforce a non-heterosexual lifestyle that affirms a new identity and provides a new means of belonging”. In similar studies on the general LGBTQ+ population, many LGBTQ+ people still define their biological and legal relatives as members of their current family, and few report belonging to a chosen family [36]. Here, in the general LGBTQ+ population, the notion of chosen families resonates, but the acceptance of chosen family members is mostly a complementary act rather than replacing the original definitions of one’s current family. As an extension of this conversation, the design of this study was fashioned to explore the relevance of chosen family arrangements within LGBTQ+ refugee settings. This particular study is interested in understanding whether the chosen family arrangement has any type of resonance within the lives of LGBTQ+ refugees.

### 1.3. Objective of the Study

This study assumes that the conceptualization of family or chosen family is different in the population of LGBTQ+ refugees because of the mere cause of the plight inflicted by their biological families [20,22,29,36,37]. The study aims to conceptualize the meaning, construction, and characteristics of new conceptions of family after LGBTQ+ refugees have fled to a safe place and have started settling there. This study is focused on instances in which LGBTQ+ refugees considered or rejected their biological family, or assailants in some cases, and whether or not the alternatively created “chosen family” filled the void left by any loss of support from their biological family. It is also the aim of this study to critically examine how the newly constructed realities of these LGBTQ+ refugees mitigated their losses, and how much it contributed to their resilience in the resettlement process.

Methodologically, the study aims to reach a conceptualized definition of family and chosen family by LGBTQ+ refugees through a data-driven approach. The aim of the study is to extract the exact conceptualization of the definition of family and chosen family, without the inclusion of the authors’ inference as to what the definitions might be. Text network graph analysis was used to extract important words and topical clusters within the narrations of the LGBTQ+ refugees. Using the concept graph created by the text network analysis, analytical meaning was extracted from the connection between words and important topic clusters.

Throughout the process of the study, the conceptualized definition of what family and chosen family is for LGBTQ+ refugees was intended to add knowledge to the gap by showing the types and characteristics of relationships. This analysis is likely to inform innovative interventions that are responding to LGBTQ+ refugee’s wellbeing, social inclusion, and sense of community.

## 2. Materials and Methods

### 2.1. Sample

Our sample was extracted from YouTube videos featuring LGBTQ+ refugees. Our initial sample included a total of 332 video clips. These video clips were retrieved from search inputs of four key phrases that were entered into YouTube’s search engine in a general-to-specific order, namely: (1) “LGBTQ+ refugees”, (2) “chosen families”, (3) “LGBTQ+ refugees’ family”, and (4) “LGBTQ+ refugees” definition of the family.

YouTube displayed 6310 outcomes when the phrase “LGBTQ + refugees” was entered, but we were only able to watch and assess 720 of the 6310 video clips based on the relevancy of the searched term. When the phrase “LGBTQ + refugees” was entered together with “chosen families”, YouTube displayed 200 relevant videos. In addition, 70 and 62 relevant videos were displayed for the search phrases “LGBTQ+ refugees family” and “LGBTQ+ refugees” definition of the family, respectively. In these searches, there was always a disparity between the available videos and the accessible videos. YouTube provides a synthetic estimate of searched results, only a fraction of which are retrievable [38]. Therefore, after retrieving a total of 332 initial outcomes using the above phrases, relevancy assessment, duplicate exclusion, and inclusion and exclusion criteria application, this study finally prepared 82 videos from 68 LGBTQ+ refugee narrations.

This study utilized video content that featured LGBTQ+ refugees in an interview setting. We excluded videos that were created for the purpose of a public campaign. In this study, statements that were directly given by the LGBTQ+ refugees were carefully collected. Using YouTube videos for text network analysis in this fashion enabled the identification of subjective words and statements. In this study, there was an absolute necessity to infer the definition of family and chosen family solely based on the exact depiction and world view of the LGBTQ+ refugees studied. More often than not, studies on the experience of individuals are skewed to the interpretation of researchers. Here, using text network analysis by extracting the individuals’ exact depiction and inference on the studied issue allowed for delving more deeply into the data, decreasing the chance of subjective interpretation bias.

Locating and recruiting diverse LGBTQ+ refugees was also a difficult and time-consuming task. The availability of videos that are potentially relevant to the issue was another reason for using YouTube videos. All of the video contents used for the analysis were created in the English language. In other words, all of the statements given by the LGBTQ+ refugees were given in English. The relevancy assessment and the criteria for the inclusion and exclusion of the samples are described in detail in the following sub-sections.

### 2.2. Relevancy Assessment

The relevancy of the videos for text network analysis was assessed by the content, the authenticity of the video uploader, and a clear statement by the LGBTQ+ refugee about the term family or chosen family. The analysis included videos with contents of interview settings that featured the refugees talking about their migration experience as part of their relationship with their families. The authenticity of the content creators or the video uploaders was also assessed by checking the consistency of the content on their channel as well as the legitimacy of the uploader as an organization or as an institute. In addition, the relevant videos were videos in which the refugees explicitly defined or gave a statement defining what family or chosen family means to them.

The initial selection of the videos and their verification for relevance was conducted by two research investigators. The initially selected videos were verified for relevancy by the primary research investigator. These selected videos were filtered and assessed for relevancy by the other research investigator.

The selection of transcribed tests and statements from the videos was also performed in the same fashion.

### 2.3. Inclusion and Exclusion Criteria

After the videos were selected and downloaded based on their titles, the content of the videos was segregated based on the following inclusion and exclusion criteria (Table 1).

### 2.4. Analysis Strategy: Concept Generating Using Text Network Analysis and Data Analysis

The first part of the data analysis entailed collecting the captions and transcribing the selected videos. Within the videos, LGBTQ+ refugees shared their personal experiences of acceptance, rejection, or exclusion in their families, as well as what family means to them. They also shared what it means to be part of a chosen family in light of psychological and physical gains and losses.

This study used InfraNodus (Nodus Labs, Leeds, UK) an insight generating tool that performs a text network analysis [39]. The tool uses a method and algorithm for identifying the pathways for meaning circulation within a text [40]. The analysis is done by visualizing the transcribed textual data as a graph, and deriving the key metrics for the concepts and the text as a whole using network analysis [40].

The resulting transcribed text data and graph representation were used to detect the key concepts about what family is or what words are associated with it. Based on the text network analysis algorithm, the analysis represented the transcribed data as a network and identified the most influential words in the data discourse based on the word co-occurrence. The graph community detection algorithm is also applied by the tool to identify the different topical clusters, which represent the main topics in the text as well as the relations between them. Finally, to make sense of the identified words, a thematic analysis was used to understand and conceptualize meanings of the words that appeared in the text network graph analysis. The influential words and topical clusters were reexamined as exact representations, and as interpretations of LGBTQ+ refugee experiences and their social worlds [41,42,43]. Based on the results rendered, a thematic analysis was performed only to interpret or make sense of the most influential words within the context of the LGBTQ+ refugees’ statements. In the thematic analysis for identifying patterns and meaning, the researchers returned to the raw data to understand the contextual meaning inferences of the words that appeared to be influential.

#### How InfraNodus Tool Perform Text Network Analysis

By utilizing the InfraNodus tool, the role of the researcher was to feed the raw textual statements to the tool in their original form. In doing so, proper care was employed by the researchers so as not to tamper with the inference of the statement, which might eventually distort the conceptualization process. Thus, the captions acquired from the YouTube videos were fed to the tool in their original form. Generally, most of the statements given by the LGBTQ+ refugees were on the migration experience as an LGBTQ+ refugee, their experience within or without a family, and their settlement process within their chosen family. To understand the meaning attribution of what family and chosen family within the statements given by LGBTQ+ refugees, a text network analysis [40] was performed on a text corpus containing every definition given. The InfraNodus tool is designed to perform text-related analysis. It is a tool that creates sense of pieces of disjointed textual data [40]. The tool visualizes the text corpus as a network; shows the most relevant topics, their relations, and the structural gaps between them; and enables an analysis of the discourse structure and the assessment of its diversity based on the community structure of the graph [39].

The tool initially removes the syntaxes and converts the words into their morphemes to reduce redundancy [39]. For example, “families” turn into “family” and “communities” becomes “community”. Next, the tool removes articles, conjunctions, auxiliary verbs, and some other stopwords, such as “he”, “she”, or “the”. Then, the remaining converted words are displayed as a network graph, where the nodes are the different words, while the edges represent their co-occurrences. The tool highlight nodes (words) that appear together the most and detect topical groups of nodes (words) that tend to appear together more often. The final result of the analysis is a visual network graph representation of the text. Based on colors and sizes, the tool enables a clear vision of the text data structure and its occurring topics [39]. In our text network analysis, the focus was thus on collecting and analyzing the statements given about the migration experience as an LGBTQ+ refugee, their experience within or without a family, and their settlement process within a chosen family.

### 2.5. Justification for Using Youtube Videos for Analysis

As noted by many researchers, the available diverse video content on YouTube is becoming an extensive and vast database. These contents serve as means for understanding the world through the eyes of the video uploaders [44,45,46]. However, to this day, there is no consensus among researchers on how to ethically use and evaluate these types of data that are available on social media platforms [47,48]. When convenient, some researchers mention obtaining permission from the uploaders to use their videos in their research [49,50,51]. Some also mention how they dealt with the ethical issues and why it is appropriate to use such data [52,53,54]. The justification given by the researchers to use such videos is that these videos are automatically in the public domain as soon as they are uploaded. In other words, while uploaders of the video can always opt to block anyone to access their video, anyone who is uploading the videos is uploading them with the understanding that the videos can be accessed by anyone with internet access.

As authors of this study, like some other researchers, we also had to struggle with the unsettling uncertainties that exist when utilizing such data [55,56,57,58,59,60,61]. However, based on the content provided in the videos and the actual information utilized in the analysis, the manner of our usage is justifiable based on the following points:About informed consent: acquiring consent from everyone featured in the video contents would be close to impossible. Permission to use the videos was unrestricted; however, consent was deemed to be given for the use of the contents. Proper responsibility was taken to ensure the confidentiality of those creating the video.The unique research opportunity: given the best input and analytical potential that these video contents provide to study the issues of LGBTQ+ refugees, the decision to use the data was also fairly justified. As a unique and very often hidden vulnerable group, having such data for gathering insight into their world was also considered for its research result value.About privacy: identifying personal information such as names, location, and any implicating identifiers were purposefully avoided from consideration in the analysis of the study. Individual identifiers were less considered and the analysis was focused on understanding the common characteristics of the LGBTQ+ refugees as a group.

### 2.6. Data Sample

The individual count and the special characteristics within the sampled video narrations from LGBTQ+ refugees are shown in Table 2. The study captured the 68 LGBTQ+ refugee narrations within 82 videos.

## 3. Results

### 3.1. Establishing a Concept Graph

As per the objective of the study, the first stage of the analysis was establishing a concept graph, constructed using narrative text inputs from the statements given by LGBTQ+ refugees. Every reference to the family within the transcribed text captions was gathered to form the concept graph. The collected text reference to family ranged from an explicit one statement definition of who is considered family to relationship qualities within the scope of what family is.

The term family is described in a varies forms of expressions. Important words used to conceptualize family and the connections between words are presented in Figure 1. Some of the words identify specific people or types of people who should be included in the definition of a family (people, mom, dad, sibling, brother, friend, community, partner, etc.), while others have offered descriptive words of the quality of family life and the relationships that describe family (love, team, happy, belong, difficult, understanding, safety, etc.). Such types of topical aggregations are not a definitive method of conceptualization used by the refugees; many of them associate people and qualities when describing family. For instance, one definition stated the following: “…I think family is to be together with someone you love… it’s where you feel safe. Family can be formed with you and your parents and your brothers and sister and also it can be formed with you and your lover and even can be formed with you and your pet... it can be with anything you love” (Gay refugee).

Within the concept graph, the biggest nodes, i.e., love, people, parent, #LGBT#community, home, life, indicate the scope and intricate elements used by LGBTQ+ refugees to define family and chosen family. The node #LGBT#community is the combination of two separate words that are separately used to express a distinct and separate idea. After examining the raw data, the use of the two words in the concept graph were combined and were used together 89% of the time by the LGBTQ+ community.

With the concept graph alone, it is safe to demarcate the ideal scope of the family as something that revolves around people who could be parents or anyone; to always involve affection of love; might possibly need an alternate community, like the LGBTQ+ community, for establishment or existence to require a home setting; and that it is an integral part of one’s life. Words associated with sentiment indicate the family is where someone feels safe, loved, protected, supported, belonged, happy, and comfortable. On the other hand, based on past experiences of LGBTQ+ refugees, words with negative sentiments are also mentioned, i.e., difficult, kill, struggle, and longing (yearning).

Chosen family is mentioned as being as equally important as the original biological family. The major identified characteristics of chosen family are stated in the following snippets of participant’s narrations. 

“…In terms of finding or finding the optimal family, then, it would be probably a combination of both the original family, the one you’re born into and the one that you’ve chosen or that you even establish or want to establish. But, I think in the majority of cases it’s the chosen family that helps you then to reconnect to the family you’re born into and then maybe merge the two of them”(Lesbian refugee) 

“…Alternative family for me is extremely important. It is important to create an inclusive, safe, beautiful place in nature and with nature. Because I personally find it’s not enough to have a biological family that is one part of our life. It’s important but not sufficient. Chosen family is like a collective of like-minded people where you can have dialogue together and support each other to be able to give your shoulders to others and also put your head on others shoulder…”(Gay refugee) 

“…All the movement of migration things change the concept of family for me. It means that I have to start to discover or develop a new form of family. It means a lot to me the process of finding people that I can call father, I can call mother, or brother is very personal to me that when I found them it goes deep into my soul. This is because I was rejected by my own family right from childhood. So, it was a very difficult relationship that I have had with my parents and my siblings. So, finding people who welcomed me who loved me for being gay or for being open… um it was just overwhelming” (Gay refugee) 

“…I have a new family kind of I lost some members of my biological family or the family I used to have. But right now I have my chosen family: my husband, my partner. So, it’s like I have my friends, my network, my LGBT community. We are starting a new network. Home is where you feel secured where you feel loved where you can be legally protected...” (Gay refugee) 

### 3.2. Topical Groups and Influential Discourse Elements 

The main topical groups of the narrations were also extracted from the concept graph. According to the topical essence of the graph, about 70% of the topical group revolves around the words #LGBT#community, home, and important (26%); love, understanding, and kind (18%); people, life, and find (17%); and parent, father, and mother (11%; see Figure 2).

When exploring the sentences that arose out of the main topical groups, family or chosen families were not necessarily associated with biological families, but family was associated with affectionate feelings about someone, connections to a community, and mutual love and understanding. The biggest node on the concept graph, i.e., people, was conceptually associated with an inferred meaning of “anyone who is like-minded, understanding, trusting, welcoming, loving, and committed”. In some instances, biological families were mentioned as being an integral part of the definition of family. However, the term biological families was situated in the same topical group as partner, friend, and #LGBT#community, which signifies the equally important roles of non-biological relationships.

The conceptualization of biological family was associated with not sufficient, got lost, need to understand more, biological kinship not necessary to establish a family, and not important. The implications of these ideal conceptualizations were noted in a significant amount within individual narrations. When choosing between typical representative statements, the following snippets portray the distinction between the importance, the role of biological and non-biological families, and the core word associations.

“…It is important to create an inclusive, safe, beautiful place in nature and with nature. I personally find it not enough to have a biological family. That is one part of our life. It’s important, but not sufficient” (Gay refugee)

“…I have a new family…I kind of I lost some members of my biological family or the family I used to have…but right now I have my chosen family my husband my partner. I have my friends, my network, my LGBT community. We are starting a new network” (Gay refugee) 

“…The thing is, with my biological family, my original family, I still love them but I need them just to understand me…” (Bisexual refugee) 

“…Family to me is the core of someone’s life. It’s where someone feels they belong, where someone feels they’re comfortable, it’s where someone feels they can trust their lives and not worry about anything. So, it might not be biological, it might be circumstantial depending on the situation where you are” (Gay refugee) 

“…I think family for me means something that’s not connected to this to the biological sphere. So, I strongly believe that the families that are really important are the families that we make as we go along and families that exist outside the prescribed bounds of kinship” (Lesbian refugee) 

“…Family goes beyond biological and I consider my partner my biological family and my closest friends to be my family” (Gay refugee) 

The influential elements within the discourse of the narrations are stated in Figure 3. With relatively higher betweenness centrality [58], the first five influential words (love, people, parent, #LGBT#community, and home) indicate topical ideas used in the discourse of the participant narrations. Betweenness centrality is a way of detecting the amount of influence a node has over the flow of information in a graph. It is often used to find nodes that serve as a bridge from one part of a graph to another [58]. According to interpretations of betweenness centrality, these nodes with high betweenness centrality have considerable influence within the concept graph because they control over information passing between other nodes [58]. Within these influential discourse elements, the reality as LGBTQ+ refugee as well as their conceptualization of what family is weighing in the words of an integral part of the narrative discourse. Especially, the top influential nodes solidify the involvement of people other than a biological relationship as an essential part of the definition of Family. LGBT communities and people are mentioned providing an alternative safe haven in the process of settlement and inferred as an integral part of the newly constructed conceptualization of what family is.

As belongingness to a certain community or being included in a home is a universal human condition, the connection of the words around the word “home” indicated how the sampled LGBTQ+ refugees all longed to be home but as it is in many countries, it is very difficult for most of them to have that sense of home. The safety of home and the sense of belonging to a certain community cease to exist during and after their flight since it is their families that abandon them forced them out of their communities. These newly established communities thus provide not only fortitude but a sense of belonging and a new start.

In addition, the frequency (the number of word occurrences) and degree (the number of connections it has to other words) of the concept graph were analyzed (see Figure 4). The top repetitively occurring words and the highly connected words indicate the intricate relationship of feeling, the person, and the space needs associating with the LGBTQ+ refugees’ conceptualization of what family is. In the frequency analysis, it was also indicated that people and friend, or non-biological ties, had an essential position in the definition of family by LGBTQ+ refugees. The biological familial ties for the terms parent, father, brother, and mother, were mentioned as still having an important position in the definition of family. However, it is also important to indicate that in some instances, these words were mentioned with a negative connotation regarding the meaning of family. Resentment, fear, and need for understanding were some of the ideal connections mentioned along with these words. For example:

“…I was chased away from home by my family, because my father said I’m terrible. I will spoil his children” (Gay refugee) 

“…Parents always have an image of what they want from the kids and I can imagine that there are fathers and mothers who don’t know how to deal with a gay, lesbian, transgender son” (Bisexual refugee)

“…My brother is embarrassed of me because I brought shame on him” (Lesbian Refugee) 

“…My dad caught me. He like walked in on us and take me downstairs and my brother was there. They took me to the store we have and made a big show out of me. A stick breaks on my hand and then he takes a metal rod. And he started beating me with it and he asked me brother to give him the scissors.….my brother and my father hold me. And my dad cut my hair. And then my dad takes the charcoal and burnt my hand. This was my first time (being in a gay relationship). Yeah and he promised if I do this again, I will be killed. Now if I write to them (know where I am) and I know my dad, he promised he would kill me and I know when he promised he would do it” (Gay refugee) 

## 4. Discussion

According to the conceptualization derived from the results of the study, a family is identified by specific people or types of people, who are or should be included in the definition of family. The result also indicates the necessity of emotional ties among the people involved. This established familial relationship with “people” as contested with biological kinship ties also implicates the initial need for the establishment of chosen family arrangements. The conceptual flow of defining family in the LGBTQ+ refugee setting highly and explicitly incorporates the concept of choice or independent construction of the ideal family in an alternate home setting.

The conceptualization of family in the concept graph resonates with the study by Weeks et al. [4]. Characteristically, chosen families are established as a “positive step to underpin a non-heterosexual lifestyle, which both affirms this identity and provides a new way of ‘belonging’ in the social world”. In another words, the LGBTQ+ refugees in this study, by adhering to alternative family settings, are affirming their sexual identity as well as maintaining a sense of belonging to a certain community.

Sentimental word associations indicate the need and the fear within familial relationships. The post-migration experience of the refugees, where the family is the assailant, coupled with settlement stress after exile, necessitates the need for safety, love, protection, support, belongingness, happiness, and comfort. For the most part, the negative-sentiment words affirm the assailant position of biological families.

Equally as important as the original biological family, chosen family is mentioned as a safety net in the settlement process. As it is mentioned in studies of the general LGBTQ+ population, more than heterosexuals, alienation from their biological family relationships is common, so friendships and alternative relationships are more critical for the LGBTQ+ individual [6,59,60]. For the LGBTQ+ refugees, a similar challenge is applicable along with the additional challenges of establishing new ties and social belongingness. Studies on chosen families [35] and their complex and robust social support networks indicate its resourcefulness toward eliminating the harmful impact of stigma and homophobia for these sexual minorities [61].

In this study, the main topical clusters affirm the shift of meaning from the traditional definitions of family. It also indicates that alternative family settings in the LGBTQ+ refugees’ circumstances are not a matter of option, but of necessity. Words within the topical groups imply an inadequate emotional transaction with biological family, while chosen family members are highlighted as people who support, love, are close, care for or nurture, reciprocate, make effort, share experiences, accept, and are trustworthy. In such instances, chosen family settings not only mitigate the needs of the refugees by assisting in the process of healing past traumas, but also provide a solution for the present needs of resettlement, acculturation, and other related needs.

The top repetitively occurring words, highly connected words, and most influential words within the conceptual graph not only indicate the conceptual shift within the definition of “family”, but could also inform innovative interventions that respond to the LGBTQ+ refugees’ need for a family. The influential words highlighted people other than those of a biological relationships, alternative communities (LGBTQ+ communities), and belonging to a home (or certain inclusive community) as essential.

Based on the results of the study, there should be an academic need and a service provision need to understand LGBTQ+ refugees as a more vulnerable type of minority. Considering the impact chosen families have on the wellbeing of different types of sexual minorities, there are no studies conducted on the issue of chosen families responding to the needs of LGBTQ+ refugees. Even to this day, sexual identities in the general public are generally viewed as anti-social or outside the mainstream; therefore, being labeled an LGBTQ+ person is often found to go against the status quo of what family is, and these people are often reduced to individuals without a home or a family [35]. In this regard, the plight of LGBTQ+ refugees apparently will double, because most of them definitely lost not only lost their social ties when they fled their country, but had to also endure resettlement stress, acculturation stress, and isolation stress.

Again, understanding the potential of chosen families in light of the LGBTQ+ refugee experience helps practitioners empathize and provide appropriate and fitting treatment within or through chosen families. When dealing with this group, the practitioner will be able to consider the perspective of the LGBTQ+ refugee in the wider social, cultural, and legal facets of wellbeing, and accommodate solutions within the settings of chosen families. The intervention within “chosen families” will thus have two roles. First, it will help navigate the traumatic past by assisting the recovery process. Second, it prepares the refugee for his/her new life by creating positive and enabling social connections.

Within the bigger framework of protecting sexual minorities and from a practical point of view, the extent to which the chosen family can be viewed as a metaphorical or concrete word needs to be further discussed. Considering its advantage in the wellbeing and settlement process of LGBTQ+ refugees, policy considerations for the establishment and promotion of chosen families for LGBTQ+ refugees should also be the assignment of refugee-hosting countries.

## 5. Conclusions

More often than not, the biological family is the catalyst that pushes an LGBTQ+ refugee to flee. This study, in a totally data driven fashion, conceptualized the meaning attributed to family and chosen family through the narrations of a number of LGBTQ+ refugees. This study also helped to clarify the meaning, significance, and role of a chosen family in the newly established lives of the refugees, by highlighting its value of safekeeping the wellbeing and settlement process.

The meaning circulation analysis in the form of text network analysis produced an intricate, but single, concept of what family or chosen families mean to LGBTQ+ refugees. The conceptualization fashioned an ideal deduction within family relationships, exploring why people other than biological relationship appear vital in their overall wellbeing and settlement, and the process through which it occurs.

According to the analysis of this study, the ideal association with biological family is expressed with words that instigate fear, danger, difficulty, and insecurity; while the ideal associations with chosen family are expressed with words like trusting, like-minded, understanding, welcoming, loving, committed, etc. This particular finding affirms the typical shift of LGBTQ+ refugees from kinship-based familial relationships, transforming to an alternative form that relies on people, anyone, or anything. This transformation by itself could be a source of strength or a soft spot exposing vulnerability. Thus, promoting, empowering, or institutionalizing alternative families within interventions for LGBTQ+ refugees is not only an appropriate measure, but should be a necessary integral part of intervention that aims to alleviate various hardships facing such refugees post migration.

The strength of this particular study is in utilizing an original research method that is practical for such studies, which is interested in insight and concept generation. This particular study made use of a potential data source, YouTube, which shares the contents of marginalized and hard to access groups such as LGBTQ+ refugees. The result of the study also provided a totally data driven result that depicted the subjective view of LGBTQ+ refugees about family and chosen family. This study is also an initial call for the relevant research community to produce more evidence in such settings, and to encourage anyone to refine the concept and to use it.

## 6. Limitations of the Study 

The lack of uniformity in the collected video contents should be mentioned as a limitation for this kind of study. Available structured and more uniformed video content on the issue that the study analyzed might have resulted in a more concrete and more enhanced conceptualization and meaning attribution for family or chosen family within the LGBTQ+ refugee setting.

There is also a study bias of only including English videos in the analysis. At the moment, the program used for analysis, InfraNodus, is only efficient to analyze samples in English, German, and Russian. We assume there are non-English refugee samples that would likely to contribute to the conceptualization process, if only their samples could be assessed for relevancy.

## Figures and Tables

**Figure 1 healthcare-09-00369-f001:**
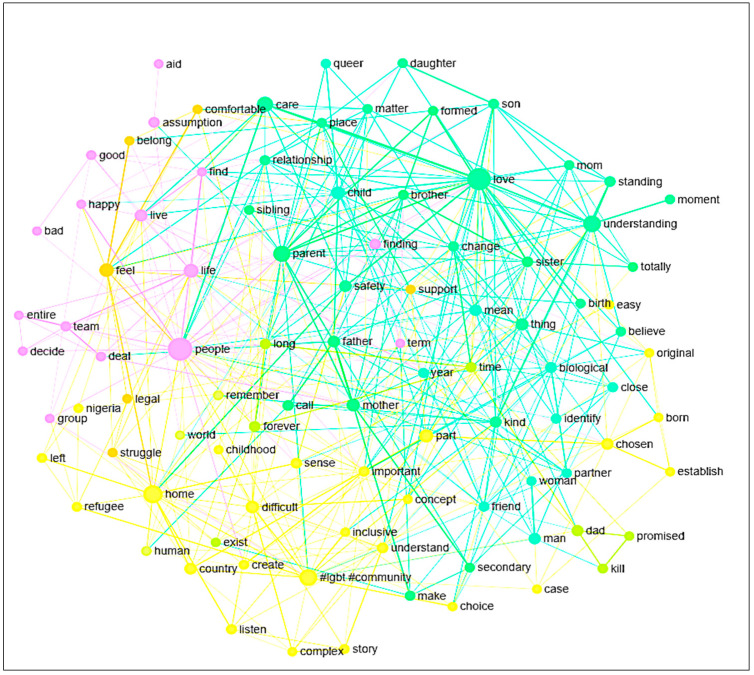
Concept graph.

**Figure 2 healthcare-09-00369-f002:**
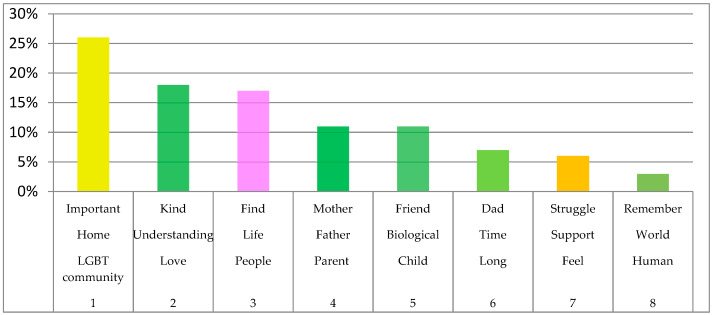
Main topical groups.

**Figure 3 healthcare-09-00369-f003:**
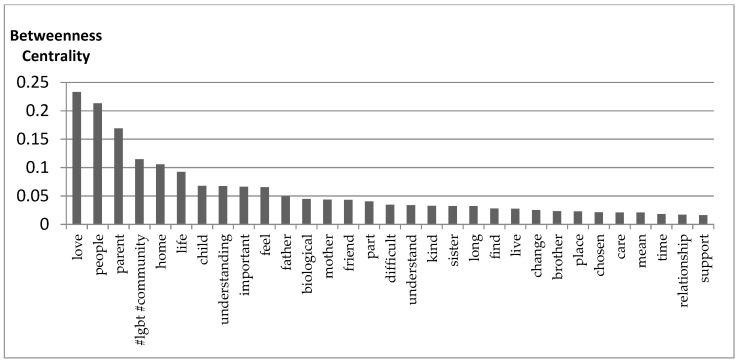
Influential discourse elements.

**Figure 4 healthcare-09-00369-f004:**
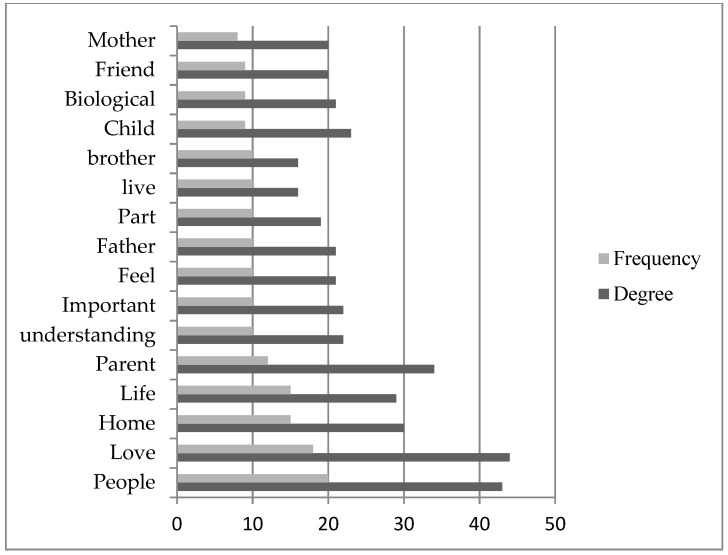
Degree and frequency of influential words.

**Table 1 healthcare-09-00369-t001:** Relevancy assessment and inclusion and exclusion process.

Video Content Relevancy Assessment	Inclusion Criteria	Exclusion Criteria
Must be a LGBTQ+ refugee content	Videos with testimonial accounts of LGBTQ+ refugee experience and relationship with biological family	no LGBTQ+ refugee experience
The uploader must be authentic and consistent	Videos containing LGBTQ+ refugee’s personal definitions of “family” in a LGBTQ+ context	No clear definition to “family”
	Videos containing LGBTQ+ refugee’s personal definitions of “Chosen family” in a LGBTQ+ context	No clear definition to “chosen family”
	Videos with English captions or videos that could be transcribed into English manually	Videos without clear statements in English

**Table 2 healthcare-09-00369-t002:** Participant characteristics.

LGBTQ+ Refugees (*N* = 68)								
Sexual Orientation *n* (%)				Gender Identity *n* (%)			Queer Identity *n* (%)	
Gay (Male)	Lesbians	Bisexuals	Other/Questioning	Male	Female	Transgender or Gender Queer	Queer	Non-Queer
25 (17.2)	22 (15.1)	16 (11)	10 (6.9)	25 (17.2)	29 (20)	12 (8.2)	22 (15.8)	15 (10.3)

## Data Availability

The data presented in this study are available on request from the corresponding author.

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
