# Peer review of "Conceptualizing “Family” and the Role of “Chosen Family” within the LGBTQ+ Refugee Community: A Text Network Graph Analysis"

_healthcare, 2021, doi:10.3390/healthcare9040369_

Round 1
Reviewer 1 Report
This paper addresses an important and under-studied topic with a thoughtful methodology. The policy implications of this study are important, and the paper merits publication. As a whole, the research is sound, but the paper would benefit from some clarifications and revisions.
- Some editing for syntax is necessary.
- The introduction needs to define the LGBTQ+ acronym (e.g., what the letters stand for, and perhaps further explain).
- Page 2, first paragraph – this sentence needs to be clarified: “In the LBGTQ+ population, the profound impact of family in the life of LGBTQ+ individuals is mentioned rather than explored further.”
- Page 2, second paragraph, last sentence – a closing quotation mark appears to be missing (unless the open quotation mark is a typo).
- Page 2, third paragraph, define what the UNHCR is.
- Page 3, lines 132-138 seem as though they’d be a better fit under section 1.3, as they refer to the purpose and objectives of the study.
- Section 2.1 should briefly note that the relevancy assessment and inclusion and exclusion criteria will be discussed in the sections that follow.
- From the discussion in Section 2.3, it sounds like videos were only included if they referenced both biological family and chosen family. Is this accurate? If so, does this result in a sample biased toward those narratives focusing on chosen families (that is, persons who speak only about biological family and not chosen family are excluded)? I don’t think that’s a problem given the objectives of the research, but that should be acknowledged if that’s the case (because, without a comparison, the research can’t speak to the frequency with which refugees turn to chosen families – but only to the experiences of those who do have chosen families). Also, were there any patterns among the sample in terms of which country the refugees settled in?
- Some additional description of the methodology, in section 2.4 or 2.4.1, would be helpful. Specifically, does InfraNodus group the words together automatically (as described in 2.4.1), or do the researchers specify root words that the program then utilizes as part of its search? And, some more information on the type of thematic analysis that was performed by the researchers after acquiring the InfraNodus results (e.g., what type of coding or other assessment) would be helpful.
- For Section 2.5, is this considered “human subjects research” under relevant institutional review board guidelines, or would this be exempt from that consideration due to public nature of the videos?
- For Figure 1, I’m curious as to whether there’s a specific node that is more closely linked to chosen family words, and whether or how that differs from a specific node that is more closely linked to biological family words. Is this something that can be depicted graphically?
- For Table 2, how is “queer” being defined? Also, the numbers do not all sum to 68. The percentages should be calculated within each category, as opposed to across all categories (e.g., what percentage of narratives represented each sexual orientation category; what percent represented each gender category; and what percent represented each queer identity category).
- Page 10, line 405 – please define what the measure of “betweenness centrality” means. I’m also unclear on why Figure 3 is a line graph as opposed to a table or bar graph – that is, why or how the individual terms are “connected” through the line depicted.
- It looks like some words are missing on Figure 4 – there are 16 sets of lines (32 lines total), but only 8 terms are listed.
- Page 12, line 475 notes that “For the most part, the negative sentimental words affirmed the assailant position of biological families.” Is there any metric from the data that indicates the relative frequency of positive versus negative assessments of biological families? That is, is there a difference between refugees fleeing an oppressive situation but retaining family support, versus refugees fleeing an oppressive situation that includes oppression from family? In either case, chosen families would play an important role, but would potentially differ between the two circumstances.
Reviewer 2 Report
Make sure this study has had approval or exemption from a human subjects group such as an IRB.
This is wonderful research and should be published, but there are many grammatical changes to be made first, and a few things that need clarification.
Peer Review of Healthcare - 1137940
Conceptualizing “Family” and the role of “Chosen family” 2 within the LGBTQ+ refugee community: A text network graph 3 analysis 4
See below and attached pdf.
Abstract
Line 1 “This study analyzed the meaning attributions about ‘family’ and ‘chosen family’ by 10 LGBTQ+ refugees..” Great start, but many people are unaware of what this term means. It is an acronym for those feeling from persecution, but many readers may not know this.
Lines 17 though 19 “Biological family is associated with words that instigate fear, danger, and insecurity; while the concept of “Chosen family” is associated with words like; trusting, like-minded, understanding, welcoming, loving, committed, etc..” This sounds very definitive. Would suggest changing the wording to, “Biological family is sometimes associated with …”.
Lines 22 and 23: “It is also a call for the relevant research community to produce more evidence in such settings and to encourage anyone to refine the concept as well as to use it.” What concept? This is a bit confusing. Suggest changing this to something like this, “ It is also a call for the relevant research community to produce more evidence in such settings as it is essential toward a better understanding.”
Introduction
Lines 28 and 29, “Across different cultures, traditional definitions and practices of what a nuclear family should be often disregarded or withhold space for LGBTQ+ family members.” This sentence doesn’t to make grammatical sense. Did you mean to say…
“Across different cultures, the nuclear family is traditionally defined in a way that excludes LGBTQ+ family members.”
Lines 33 and 34, “The traditional family dynamics are often contested if the LGBTQ+ had to go against the family norm” It’s always best to use person-first language, and not call the individual an “LGBTQ+.” Would re-write it this way, “The traditional family dynamics are often contested if the individual who identifies themselves as LGBTQ+ goes against the family norm.”
Lines 34 and 35 “For in-stance, in most cases, a complete family disownment will follow an LGBTQ+ individual who is from a traditional Christian and Muslim cultural background.” Some might take offense here. Many Christian families in the US would not disown a person for coming out as LGBTQ+. Would rephrase this as, “For in-stance, in some cases, a complete family disownment could follow an LGBTQ+ individual who is from a traditional Christian and Muslim cultural background.” Or you could say, “For in-stance, in China, a complete family disownment would typically follow an LGBTQ+ individual who is from a traditional Christian and Muslim cultural background.”
Lines 42 and 43. “These exclusions, on so many occasions, led many LGBTQ+ people to seek to 42 establish “alternative families” or “families of choice” that offer them the love and security 43 that they did not find from their biological families” Well written!
Lines 45 to 55, Suggest revising for grammar, “The profound impact of family in the life of LGBTQ+ individuals is often mentioned rather than explored further [5]. For instance, in the general LGBTQ+ population, parent and family rejection is strongly associated with mental health problems, substance use, and sexual risk while perceived family support is associated with better mental health and less substance use [6-14]. Nonetheless, these findings still fall short to explain: which factors contribute to resilience among LGBTQ youth with un-supportive or rejecting families; how the presence of one supportive parent can compensate for the lack of support from another parent or guardian; how the presence of a non-parental family member can compensate for the effects of unsupportive parents; how non-parental mentors improve health outcomes; etc [5].”
Lines 55 and 56, “Recent research findings also suggest that opposition to homosexuality and same-sex relationships in recent years has softened for the general LGBTQ+ population” Did you mean to say, “Recent research findings also suggest that opposition to homosexuality and same-sex 55 relationships in recent years has softened for the general population”
Lines 58 and 59, “As it will be the focus of this study, LGBTQ+ refugees who are in exile mainly due to their sexual orientation have their sub-group characteristics.” Suggest changing the wording to: “One focus of this study will be to take a closer look at the sub-characteristics of the LGBTQ+ refugees who are in exile mainly due to their sexual orientation.”
Materials and Methods
Line 62 to 64, “These challenges will thus contest the assumption that considers opposition to homosexuality and same-sex relationships has softened for all types of sexual minorities.” Not sure what you are trying to say here. Rewording suggestion “These challenges will therefore contest the assumption that there has been a softening of opposition to homosexuality and same-sex relationships for minorities”
Lines 69 to 71, “LGBTQ+ refugees thus refer to individuals who are forced to flee their country due to persecution solely against their gender identity or sexual expression that are different from the societal norms of their origin.” Not sure what you are trying to say here. Rewording suggestion: “The term, “LGBTQ+ refugees” refers to individuals who are forced to flee their country due to persecution solely due to their gender identity, or due to sexual expression that differs from their societal norms .”
Lines 73 thru 80, This is excellent! I’m learning a lot by reading this.
Lines, 81 thru 87, “During the exile and resettlement process, studies on LGBTQ+ refugees: reported these refugees felt that there was a lack of connection with their diaspora communities [21]; more or continued hostility in an exiled country [22]; the psychologically damaging burden of proof that they are members of a sexual or gender minority group [23-25]; systematic homo-bi-trans phobia and racist systems [22,26]; higher rates of refugee status refusal compared to other groups [27-28]; resettlement inadequacies like the lack of cultural competence [29], and various mental health stress due to their minority status [30].
Lines 95 and 96, “Thus fleeing from their family and that type of society usually produced a relative relief for the refugee.” This is well-written and very interesting.
Lines 107-110, “However, given the relationship dynamics that the LGBTQ+ refugees might have with their families, relevant studies investigating this gap are scarce. In light of poor family dynamics as the primary reason for LGBTQ+ refugees to flee their country, conceptualizations of what “family” is to LGBTQ+ refugees implores an academic investigation”
Lines 115 through 117, “It is this common identity and common experience that these people use to conceptualize the meaning of a family.” Strongly suggest omitting this sentence. It isn’t needed and the term, “these people” is sometime thought to sound biased.
Lines 124 to 125, “In Weeks et al., [33] [4] , chosen families are defined as highly committed and friendship-based, kin-like relationships”
Lines 125 to 126, “The typical character of these ‘chosen families’ is that they are ‘actively created as a positive step to reinforce a non-heterosexual lifestyle that affirms a new identity and provides a new means of belonging.”
Lines 139 to 147,
1.3. Objective of the study 139
This study aims to conceptualize the meaning, construction, and characteristics of newly developed perceptions of “family” after the LGBTQ+ refugees have fled to a safe place and have started settling there. This study focused on instances in which the LGBTQ+ refugees considered or rejected their biological family or the assailants in some cases, and whether or not the alternatively created “chosen family” filled the void left by any loss support from their biological family. . It is also the aim of this study to critically examine how the newly constructed realities of these LGBTQ+ refugees mitigated their losses and how much it contributed to their resilience in the resettlement process.
Lines 153-155, “Using the concept graph created by the text network analysis, analytical meaning was extracted from the connection between words and important topic clusters.”
Methods - Text network graph analysis and InfraNodus– Well Explained.
“Therefore, after retrieving a total of 332 initial outcomes using the above phrases, relevancy assessment, duplicates exclusion, and inclusion and exclusion criteria application, these researchers prepared 82 videos from 68 LGBTQ+ refugee narrations.”
Lines 181-182, “There were videos that were excluded because they were created for the purpose of a public campaign. In this study, statements that were directly given by the LGBTQ+ refugees carefully collected.”
Lines 182-183, “Using YouTube videos for Text network analysis in this fashion enabled the identification of subjective words and statements.”
Lines 187-189, “Here, text network analysis by extracting the individuals’ exact depiction and inference on the studied issue, allowed these researchers to delve more deeply into the data, decreasing the chance of subjective interpretation bias.”
Lines 190-194, “Locating as well as recruiting diverse LGBTQ+ refugees is also a difficult and time-consuming task. The availability of videos that are potentially relevant to the issue is another reason to use YouTube videos. All the video content used for the analysis were created in the English language. In other words, all the statements given by the LGBTQ+ refugees were given in English.”
Table 1. This is good.
Line 216, “The first part of the data analysis entailed collection of the captions and transcribing the selected videos.”
2.4.1. How the InfraNodus tool performs Text network analysis
This section is well-written.
Lines 280-283, “About informed consent: acquiring consent from everyone featured in the video contents is close to impossible. Since permission to use the videos was unrestricted, however, consent was deemed given to use the contents. Proper responsigbility was however give to ensure confidentiality of those creating the video. 3.1. Establishing a concept graph
As per the objective of the study, the first stage of the analysis was establishing a concept graph, constructed by the narrative text inputs from the statements given by LGBTQ+ refugees…”
Lines 306-307, “Important words used 306 to conceptualize family and the connections between the words are presented in Figure 1.”
Results – This section is awesome! Wonderful data. Just watch the tenses of all verbs and stay away from the first person language.
Discussion - This section is awesome! Well worded, and nice tie in to the literature review. Well done!
- Conclusions
More often than not, the biological family is the catalyst that pushes the LGBTQ+ refugee to flee. This study, in a totally data driven fashion, conceptualized the meaning attributed to ‘family and ‘chosen family’ within the narrations of number of LGBTQ+ refugees. This study also helped to clarify the meaning, significance, and role of a “Chosen Family” in the newly established life of the refugees, by circling its value of safekeeping the wellbeing and settlement process.
Lines 543-544, “This particular finding affirms the typical shift of the LGBTQ+ refugees from kinship based familial relationships, transforming to an alternative form that is relying on ‘people’, anyone, or anything.”
